Identification of Austwickia chelonae as cause of cutaneous granuloma in endangered crocodile lizards using metataxonomics

Jiang Haiying 1 2 3
Zhang Xiujuan 2
Li Linmiao 2
Ma Jinge 2
He Nan 4
Liu Haiyang 4
Han Richou 2
Li Huiming 2
Wu Zhengjun 5
Chen Jinping chenjp@giabr.gd.cn 2
1 South China Botanical Garden, Chinese Academy of Sciences , Guangzhou , Guangdong , China
2 Guangdong Key Laboratory of Animal Conservation and Resource Utilization, Guangdong Public Laboratory of Wild Animal Conservation and Utilization, Guangdong Institute of Applied Biological Resources , Guangzhou , Guangdong , China
3 University of Chinese Academy of Sciences , Beijing , China
4 Guangdong Luokeng Shinisaurus crocodilurus National Nature Reserve , Shaoguan , Guangdong , China
5 Guangxi Key Laboratory of Rare and Endangered Animal Ecology, Guangxi Normal University , Guilin , Guangxi , China
Spilki Fernando
Electronic publication date: 2019 Mar 13
Publication date: 2019
Volume: 7
Electronic Location ID: e6574
Received 2018 Oct 2; Accepted 2019 Feb 5
Copyright: ©2019 Jiang et al.
Copyright year: 2019
Copyright holder: Jiang et al.
License: This is an open access article distributed under the terms of the Creative Commons Attribution License, which permits unrestricted use, distribution, reproduction and adaptation in any medium and for any purpose provided that it is properly attributed. For attribution, the original author(s), title, publication source (PeerJ) and either DOI or URL of the article must be cited.
License URL: https://creativecommons.org/licenses/by/4.0/

Keywords: Austwickia chelonae, Dermatosis, Lizard, Infection, Granuloma, Next-generation sequencing (NGS)

Funding: Planning Funds of Science and Technology of Guangdong Province 2016B070701016 Guangzhou Science Technology and Innovation Commission 201804020080 Guangdong Institute of Applied Biological Resources GIABR-pyjj201604 GDAS Special Project of Science and Technology Development 2019GDASYL-0105046 2018GDASCX-0107 2017GDASCX-0107 Special Funds for Forestry Development and Protection of Guangdong Province (2017) This project was supported by the Planning Funds of Science and Technology of Guangdong Province (2016B070701016), the Guangzhou Science Technology and Innovation Commission (201804020080), the Training Fund of Guangdong Institute of Applied Biological Resources for PhDs, Masters and Postdoctoral Researchers (GIABR-pyjj201604), the GDAS Special Project of Science and Technology Development (2019GDASYL-0105046, 2018GDASCX-0107 and 2017GDASCX-0107) and the Special Funds for Forestry Development and Protection of Guangdong Province (2017). The funders had no role in study design, data collection and analysis, decision to publish, or preparation of the manuscript.

==============================
The crocodile lizard (Shinisaurus crocodilurus Ahl, 1930) is an endangered reptile species, and in recent years many have died from diseases, especially the rescued and breeding individuals. However, pathogens underlying these diseases are unclear. In this study, we report our effort in rapidly identifying and isolating the pathogen that causes high mortality in crocodile lizards from Guangdong Luokeng Shinisaurus crocodilurus National Nature Reserve. The typical symptom is cutaneous granuloma in the infected crocodile lizards. Metagenomic next-generation sequencing (mNGS) is a comprehensive approach for sequence-based identification of pathogenic microbes. In this study, 16S rDNA based mNGS was used for rapid identification of pathogens, and microscopy and microbe isolation were used to confirm the results. Austwickia chelonae was identified to be the dominant pathogen in the granuloma using 16S rDNA based mNGS. Chinese skinks were used as an animal model to verify the pathogenicity of A. chelonae to fulfill Koch’s postulates. As expected, subcutaneous inoculation of A. chelonae induced granulomas in the healthy Chinese skinks and the A. chelonae was re-isolated from the induced granulomas. Therefore, A. chelonae was the primary pathogen that caused this high mortality disease, cutaneous granuloma, in crocodile lizards from Guangdong Luokeng Shinisaurus crocodilurus National Nature Reserve. Antibiotics analysis demonstrated that A. chelonae was sensitive to cephalothin, minocycline and ampicillin, but not to kanamycin, gentamicin, streptomycin and clarithromycin, suggesting a possible treatment for the infected crocodile lizards. However, surgical resection of the nodules as early as possible was recommended. This study is the first report of pathogenic analysis in crocodile lizards and provides a reference for disease control and conservations of the endangered crocodile lizards and other reptiles. In addition, this study indicated that mNGS of lesions could be used to detect the pathogens in animals with benefits in speed and convenient.

Introduction

The crocodile lizard (Shinisaurus crocodilurus Ahl, 1930) is a relict reptile and the only species of the family Shinisauridae. It is a Class I protected species in China, an endangered species on the International Union for Conservation of Nature (IUCN) Red List of Threatened Species (Nguyen, Hamilton & Ziegler, 2014), and an appendix I species by the Convention on International Trade in Endangered Species of Wild Fauna and Flora (CITES I). This species is distributed in a few isolated sites in southern China (Guangdong and Guangxi provinces) and northern Vietnam (Quang Ninh and Bac Giang provinces) (Van Schingen et al., 2014; Van Schingen et al., 2016). However, anthropogenic disturbances via resource acquisition, habitat destruction and environmental changes, among other factors, have dramatically decreased the population of crocodile lizards in the wild (Huang et al., 2008; Nguyen & Ziegler, 2015). The total number of wild crocodile lizards in China has decreased from 6000 in 1978 to approximately 1200 (Jiang et al., 2017). Recent field surveys in Vietnam also showed that the wild population of crocodile lizards in Vietnam has decreased to fewer than 150 individuals (Van Schingen et al., 2016). In addition, the lizard’s population continues to show a sharp decline.

However, during the work of rescue and breeding, the crocodile lizards are prone to serious diseases that cause many deaths each year (Jiang et al., 2017). The most frequently disease is skin diseases. For example, in 2014, sixty-nine out of about 200 crocodile lizards died mainly due to a skin disease with a typical symptom of cutaneous granuloma in Guangdong Luokeng S. crocodilurus National Nature Reserve. Similar situation was also found in Guangxi Daguishan Crocodile Lizard National Nature Reserve, another skin disease with a typical symptom of limbs ulceration and swelling causes many deaths each year. However, disease diagnosis of wildlife animals can be impeded due to the limited clinic samples and lack of information regarding pathogens that cause these diseases. While thyroid adenocarcinoma, melanomacrophage hyperplasia and suspected seizures have been described, infectious pathogens in crocodile lizards have not been previously reported (Brady et al., 2016). Rapid clinical diagnosis of infectious disease is necessary to facilitate timely therapy.

Microbial culture has been considered as the gold standard of diagnostic methods and the most widely used for bacterial and fungal pathogens, but it is time-consuming and bias for the limitations of the media utilized for growth. Rapid advances in high-throughput sequencing now make it possible to comprehensively identify the microbes in a given community, including fastidious and unculturable taxa. Therefore, in recent years, two next-generation sequencing based methods, metagenomics and metataxonomics, have been developed as a fertile area for unbiased microbial pathogenic identification and clinical diagnostics (Fukui et al., 2015; Li et al., 2018; Hilton et al., 2016; Razzauti et al., 2015; Somasekar et al., 2017). Metataxonomics is a gene marker (e.g., 16S rDNA or ITS sequence) based high-throughput microbial diversity characterization, and metagenomics is the whole genomic shotgun sequencing approach (Marchesi & Ravel, 2015). Metagenomics avoids PCR bias, and it is not restricted to only bacterial or fungal sequences, while metataxonomics can get rid of the signal from host contamination. Therefore, metataxonomics is more suitable for wildlife animals which usually lack genome sequences for mapping references. Besides, to achieve high coverage and depth for species identification, metataxonomics is much cheaper than metagenomics. However, reports on the application of metataxonomics in wildlife pathogenic identification remain rare.

This study applied metataxonomics to identify the underlying pathogens of the cutaneous granuloma disease in crocodile lizards. This skin disease caused high mortality of crocodile lizards and occurred every year in the Guangdong Luokeng S. crocodilurus National Nature Reserve. In addition, microscopy and cultivation were used to confirm the results of high-throughput sequencing. Moreover, we tried to fulfill Koch’s postulates, a scientific standard for establishing disease causation (Byrd & Segre, 2016).

Materials & Methods

Ethics statement

All experimental animal procedures in this study were approved by the Committee on the Ethics of Animal Experiments of the Guangdong Institute of Applied Biological Resources (GIABR180928 and GIABR201027) and followed basic principles.

Animals and sampling

The sick crocodile lizards were found in the Guangdong Luokeng S. crocodilurus National Nature Reserve (24°31′14″N, 113°20′18″E). All of the lizards were adults or sub-adults. They were raised in ecological simulation pools. Some of them were rescued from the local wild. The others were born and raised in captivity, and their mother or grandmother was rescued from the local wild. The Nature Reserve raised these lizards to reintroduce them to the wild.

Lesions that formed one or more nodules in the skin of the crocodile lizard were resected, as shown in Fig. 1. The skin nodules were collected and immediately stored in liquid nitrogen. Some nodules were collected from dead animals, and the others were collected by biopsies.

Figure 1 Locations of cutaneous granulomas in crocodile lizards.

The arrows indicate the lesions. One or more nodules were found under the lesion. (A) Right forelimb. (B) Right hind limb. (C) Head. (D) Under the tongue. (E) Around cloaca. (F) The right of the lower jaw. (G) Lower jaw. (H) Submental triangle. Photo credit: (A–C, E–F) Jinping Chen. (D, G–H) Nan He.

A total of 16 crocodile lizards were sampled and 33 nodules were collected in this study (Table 1). Five nodules were used for histological examination. Three nodules were used for scanning electronic microscopy (SEM). Four nodules were used for bacteriological cultivation. Eleven nodules were used for high-throughput sequencing to analyze the bacterial and fungal components of the nodules. Six soil samples and two water samples from the crocodile lizard living environment were also collected for sequencing to trace the source of the pathogen. The information of sequenced samples was provided in Table 2.

Table 1 Information of sample collection.

Animal number	Number of nodule	Location of nodules	Treatment of nodule	Sample date	
LK3	1	Under the tongue	Stored	20160524	
LK4	3	Right hind limb	DNA extraction for one nodule	20160524	
LK5	1	Under the tongue	Paraffin section	20160524	
LK0525	5	Stomach and intestinal tract	Stored	20160525	
LK6	1	Submental triangle	DNA extraction, bacteriological cultivation	20160627	
LK7	3	Not recorded	DNA extraction for one nodule	2014	
LK8	1	Around cloaca	Paraffin section	2014	
LK9	2	Not recorded	Paraffin section for one nodule	2014	
LK13	1	Head	DNA extraction	20160628	
LK16	4	Head	DNA extraction for one nodule	20160824	
LK17	2	Head, around cloaca	DNA extraction for one nodule	20160824	
LK18	3	Submental triangle	DNA extraction for one nodule	20160824	
LK19	1	Around cloaca	DNA extraction	20160824	
LK20	2	Heart	Stored	20170505	
LK21	1	The right of the lower jaw	DNA extraction, bacteriological cultivation, SEM, paraffin section	20170731	
LK22	2	Under the tongue, right forelimb	DNA extraction, bacteriological cultivation, SEM, paraffin section	20170731	

Table 2 Sample information of high-throughput sequencing.

Group	Sample	Location	Sample date	
Nodule	L.LK.07	Not recorded	2014	
Nodule	L.LK.4L	Right hind limb	20160524	
Nodule	L.LK.13	Head	20160628	
Nodule	L.LK.06	Submental triangle	20160627	
Nodule	L.LK.16	Right lower jaw	20170731	
Nodule	L.LK.17	Under the tongue	20170731	
Nodule	L.LK.18	Right forelimb	20170731	
Nodule	L.LK.19	Around cloaca	20160824	
Nodule	L.LK.20	Submental triangle	20160824	
Nodule	L.LK.21	Head	20160824	
Nodule	L.LK.22	Head	20160824	
Water	SW.LK.03	Pool where crocodile lizards with nodules were found	20160715	
Water	SW.LK.04	Pool where crocodile lizards with nodules were found	20160715	
Soil	SS.LK.03	Pool where crocodile lizards with nodules were found	20160715	
Soil	SS.LK.04	Pool where crocodile lizards with nodules were found	20160715	
Soil	SS.LK.05	Pool where crocodile lizards with nodules were found	20160824	
Soil	SS.LK.06	Pool where crocodile lizards with nodules were found	20160824	
Soil	SS.LK.07	Pool where crocodile lizards with nodules were not found	20160824	
Soil	SS.LK.08	Pool where crocodile lizards with nodules were not found	20160824	

Histology and SEM

Five skin nodules were fixed in 4% paraformaldehyde for structural observation.

For light microscopy, 2–3 µm paraffin sections were prepared and stained with hematoxylin-eosin (H&E, Hematoxylin and Eosin Staining Kit C0105; Beyotime Biotechnology, Jiangsu, China) and Grocott-Gomori’s methenamine silver (GMS, Grocott-Gomori’s Methenamine Silver Staining Kit M052; Shanghai Gefan Biotechnology Co., Ltd., Shanghai, China) stains. Slides for light microscopy were examined using EVOS® FL Auto Cell Imaging System (Thermo Fisher Scientific Inc., Shanghai, China).

For SEM, samples were dehydrated in a graded ethyl alcohol series from 20% to 100%. Subsequently, the samples were dried in a CO2 critical point dryer (Leica EM CPD300; Leica Microsystems Inc., Allendale, NJ, USA), mounted onto aluminum stubs, coated with platinum and examined under a Hitachi S-3400N SEM (Hitachi Ltd.; Japan).

Metataxonomic high-throughput sequencing and analysis

Total DNA was extracted from the skin nodules, water samples and soil samples using a PowerFecal® DNA Isolation Kit (MOBIO Laboratories, Inc., Carlsbad, CA, USA) with 20 mg/ml lysozyme. For bacterial community analysis, the V4 hypervariable region of the 16S rRNA gene was amplified with the primers 515F (5′-GTGCCAGCMGCCGCGGTAA-3′) and 806R (5′-GGACTACHVGGGTWTCTAAT-3′). The amplicon library was prepared using TruSeq® DNA PCR-Free Sample Preparation Kit for Illumina (Illumina, Inc.; USA). Sequencing on an Illumina HiSeq platform (250 bp paired-end reads) was performed by the Novogene Corporation (China). In addition, to determine whether the fungal infection was present, the fungal ITS1 sequence was amplified. The primers for ITS1 sequence amplification were ITS5-1737F (5′-GGAAGTAAAAGTCGTAACAAGG-3′) and ITS2-2043R (5′-GCTGCGTTCTTCATCGATGC-3′), and the expected size of the amplicon was 250–300 bp. 2% agarose gel electrophoresis was used to detect the amplicons.

After sequencing, raw tags were filtered using the QIIME package (Caporaso et al., 2010) to remove low-quality and chimeric sequences. Sequences with ≥97% similarity were assigned to the same operational taxonomic units (OTUs) using Uparse (Edgar, 2013). A representative sequence for each OTU was annotated using Mothur by searching the SILVA database (Threshold = 0.8) (Quast et al., 2013; Schloss et al., 2009). For comparisons between samples, the OTU abundances were normalized by the number of OTUs obtained from the sample with the lowest counts. Multiple sequence alignment and phylogenetic tree were conducted to show the general view of the sequenced bacteria based on top 100 genera using QIIME (Caporaso et al., 2010). Flower plots were drawn using R software to compare the similarities and differences in microbes among samples.

Molecular detection of ranavirus

Co-infection of A. chelonae with ranavirus was reported previously (Tamukai et al., 2016). Therefore, the presence of ranavirus in crocodile lizards was examined to confirm the pathogen. Three primer sets were used to detect the ranavirus specific major capsid protein (MCP) gene: RanaM68F (5′-GCACCACCTCTACTCTTATG-3′) and BIVMCP154 (5′-CCATCGAGCCGTTCATGATG-3′), RanaJP556F (5′-GGTTCTTCCCCTCCCATTC TTCTT-3′) and RanaJP772R (5′-GGTCATGTAGACGTTGGCCTCGAC-3′), OlT1 (5′-GACTTGGCCACTTATGAC-3′) and OlT2R (5′-GTCTCTGGAGAAGAAGAAT-3′). The expected sizes of the amplicons were 230 bp, 217 bp and 500 bp, respectively (Stöhr et al., 2013; Une et al., 2014). A total of 2% agarose gel electrophoresis was used to detect the amplicons. Samples were considered positive if two or three primer sets were positive (Tamukai et al., 2016).

Bacterial isolation and cultivation

Four granulomas were cut open and spread on Columbia blood agar base plates. The plates were placed at 30 °C for 24–48 h to cultivate the bacteria, which were then isolated and purified using repeated plate streaking. The isolated bacteria were incubated in Columbia medium. The DNA of each bacterium was extracted using the TIANamp Bacteria DNA Kit DP302 (Tiangen Biotech Co., Ltd., Beijing, China). The 16S rRNA gene was amplified with the universal primers 27Fs (5′-GAAGTCATCATGACCGTTCTGCAAGAGTTTGATCMTGGCTCAG-3′) and 1492Rs (5′-AGCAGGGTACGGATGTGCGAGCCTACGGHT ACCTTGTTACGACTT-3′), and sequenced with the primers 1S (5′-GAAGTCATCATGACC GTTCTGCA-3′) and 2RS (5′-AGCAGGGTACGGATGTGCGAGCC-3′). The bacteria were annotated by matching the similarity results in NCBI database using BlastN (https://blast.ncbi.nlm.nih.gov).

Artificial infection

Because of their endangered status and legality issues, crocodile lizards cannot be experimentally inoculated with bacteria to confirm the pathogen. Instead, another widely distributed lizard species, the Chinese skink (Plestiodon chinensis), was used to verify the pathogenicity of Austwickia chelonae and fulfill Koch’s postulates (Byrd & Segre, 2016).

All Chinese skinks were adults, captured from wild and temporarily housed in the 57 cm*42 cm *30 cm cages in the laboratory for at least one week before inoculation. The bottom of the cages was covered with grass brought back from wild or wood chips. Each cage contained three or four skinks. The cages were kept in secluded rooms to minimize human interference. The skinks were fed with 4–5 cm Zophobas morio larvae and autodrinker.

The candidate pathogen, A. chelonae LK16-18, was incubated with Columbia medium for 48 h at 30 °C and re-suspended in sterile phosphate buffer saline (PBS, pH =7.4) at a concentration of 2.67 ×108 CFU/ml. Ten experimental Chinese skinks were randomly assigned into two groups: the treated group (N = 5) and the negative control group (N = 5). The average weight and snout-vent length of the experimental animals was 26.65 g and 9.57 cm, respectively. All of the Chinese skinks were clinically healthy, and no nodules or other lesions were found before inoculation. After local disinfection with 75% ethanol, the treated group were subcutaneously inoculated with A. chelonae suspension. The inoculation dose was 20 µl/g (volume of bacterial suspension: weight of animal): 200 µl/30g for the left side of the trunk and tail, 100 µl/30g for the left forelimb and hind limb. The right lateral skin was used to comparative observation. The negative control group were hypodermically inoculated with the same dose of sterile PBS. The disease signs development was observed every day after inoculation for one month or until the experimental Chinese skinks died. When the animals died, the nodules were collected and examined for the presence of A. chelonae. The detection methods included 16S rDNA sequencing using the primers 27Fs and 1492Rs, H&E staining and bacterial isolation as described above.

Moreover, isolated bacteria corresponding to the other three common OTUs in crocodile lizard nodules (Salmonella sp., Acinetobacter sp., Pseudomonas sp.) were also subcutaneously inoculated into Chinese skinks to confirm the pathogeny of A. chelonae. Twelve Chinese skinks were randomly assigned into four groups, subcutaneously inoculated with Salmonella enterica LK18-19, Acinetobacter sp. Exi5-53, Pseudomonas protegens Exi5-13, and PBS, respectively, at the same dose and with the same method mentioned above. The average weight and snout-vent length of these skinks was 36.95 g and 10.78 cm, respectively.

Antibiotic sensitivity test

A standard disk diffusion test was used to test the antibiotic sensitivity of the candidate pathogen A. chelonae LK16-18. The bacterial suspension was spread onto Columbia blood agar base plates, and disks containing antibiotics (Hangzhou Microbial Reagent Co., Ltd., Hangzhou, China) were sterilely placed on these plates. The plates were incubated at 30 °C for 48 h, and the disk diffusion zone diameters were recorded. The sensitivity categories were interpreted according to the breakpoints provided in manufactural instruction and the Clinical and Laboratory Standards Institute (CLSI) document M100 (28th Edition) (CLSI, 2018). The antibiotic information was listed in Table 3.

Table 3 Sensitivities of Austwickia chelonae to antibiotics.

No.	Antibiotics	Disk content (µg)	Zone diameter (mm)	Interpretive categories	Interpretive categories and zone diameter breakpoints (mm)	
					S	I	R	
1	Cephalothin	30	68	S	⩾18	15–17	⩽14	
2	Ampicillin	10	48	S	⩾17	14–16	⩽13	
3	Minocycline	30	42	S	⩾19	15–18	⩽14	
4	Levofloxacin	5	37	S	⩾17	14–16	⩽13	
5	Rifampicin	5	34	S	⩾20	17–19	⩽16	
6	Erythromycin	15	29	S	⩾23	14–22	⩽13	
7	Ciprofloxacin	5	23	S	⩾21	16–20	⩽15	
8	Piperacillin	100	32	S	⩾21	18–20	⩽17	
9	Kanamycin	30	–	R	⩾18	14–17	⩽13	
10	Gentamicin	10	–	R	⩾15	13–14	⩽12	
11	Streptomycin	10	–	R	⩾15	12–14	⩽11	
12	Clarithromycin	15	–	R	⩾18	14–17	⩽13	

Sequencing data availability

All raw sequences obtained from high-throughput sequencing were deposited into the NCBI Sequence Read Archive (SRA) under the accession number SRP152217. The complete genome sequence of A. chelonae LK16-18 has been deposited into NCBI GenBank and published (Jiang et al., 2018). The 16S rRNA gene sequences of A. chelonae LK16-18, S. enterica LK18-19, Acinetobacter sp. Exi5-53 and P. protegens Exi5-13 have been deposited into NCBI NR database under the accession numbers MK110377, MK235186, MK235211 and MK235212.

Results

General description of the disease

This disease was characterized by one or more nodules (Fig. 1). The clinical signs were located on the head, limbs and tail but not on the dorsal or ventral skin (Fig. 1). The most common nodule location was under the tongue (Fig. 1D). Usually, the nodules were located in the skin system. However, there are two exceptions. One lizard was found to have nodules in the stomach and intestinal tract. The other lizard was found to have nodules in the heart (Table 1).

Usually, one nodule was located inside one lesion. Two or more nodules were rarely found, but some crocodile lizards with two or more lesions were observed. The nodules were light yellow or white (Fig. 2). Some advanced nodules were caseous necrotic.

Figure 2 Nodules in the lesions of crocodile lizards.

Photo credit: Jinping Chen.

No ectoparasites were found on the surface of lesions.

Results of histology and SEM

Five nodules used for histological staining showed the same characteristics. Microscopic images at low magnification showed that the nodules were histologically caseous necrotic with a membrane outside the nodule and layer structure inside the nodule (Figs. 3A, 3C). A lot of filamentous bacteria were observed inside the nodule (Figs. 3B, 3D). The filamentous bacteria occupied the whole nodule. Some mycelia have penetrated into the membrane (Fig. 3D). Scanning electron microscopy also showed a large number of filamentous bacteria across the necrotic tissue in the nodule (Fig. 4).

Figure 3 Histological micrographs of nodules.

(A) The multi-layered structure of the nodule. H&E staining. Scale bar = 400 µm. (B) A high number of filamentous bacteria were stained in dark blue inside the nodule. H&E staining. Scale bar = 100 µm. (C) The membrane and multi-layered structure of the nodule. GMS staining. Scale bar = 400 µm. (D) Filamentous bacteria were stained in black. GMS staining. Scale bar = 100 µm.

Figure 4 SEM image of the inside of a nodule.

Scale bar = 15 µm.

Detection of bacteria

16S rDNA based metataxonomic high-throughput sequencing was used to determine the bacterial compositions of 19 samples, including 11 nodules from the skin of crocodile lizards and 8 environmental samples (Table 2). Each sample contained at least 31,635 effective sequences (Fig. S1). The rarefaction curves showed that these sequencing depths were sufficient for capturing microbiota in each sample, especially in the nodule samples (Fig. S2). A total of 54 phyla were sequenced in this study. The phylogenetic relationship of top 100 genera was showed in Fig. S3.

The flower plots showed that five OTUs were common to all nodule samples: Austwickia chelonae, Salmonella sp., Acinetobacter sp., Pseudomonas sp. and Halomonas sp. (Fig. 5). Relative abundance analysis showed that these five OTUs were predominant, with relative abundances of 29.0%, 8.7%, 1.0%, 0.3% and 0.2%, respectively, in the total dataset of nodules. In addition to the five OTUs mentioned above, the dominant bacteria in nodules also included Bacillus, Fusobacterium, Bacteroides, Chryseobacterium and Morganella (Fig. 6). However, these genera were not present in every nodule.

Figure 5 Flower plot conducted based on OTUs.

The core number in the middle represents the number of OTUs common to all samples. The numbers on the petals represent the number of OTUs unique to the sample.

Figure 6 Relative abundances of nodule and environmental bacteria at the general level.

According to Koch’s postulates, the pathogen must occur in every case of the disease (Byrd & Segre, 2016). Therefore, the five common bacteria, A. chelonae, Salmonella sp., Acinetobacter sp., Pseudomonas sp. and Halomonas sp., were identified as the candidate pathogens underlying nodules in crocodile lizards. Of these five bacteria, only A. chelonae was filamentous, which was consistent with the morphology observed in H&E staining, GMS staining and SEM (Figs. 3–4), while the others were rod-shaped. These results suggested that A. chelonae was a candidate pathogen for this crocodile lizard disease.

Detection of fungi

For fungal infection analysis, only two out of 11 nodule samples showed a clear band, and three samples showed a weak band at the expected size of the ITS1 sequence (Fig. 7). In other words, not all of the nodules contained fungi. Moreover, no fungi were observed at GMS staining and SEM micrographs.

Figure 7 Agarose gel (2%) electrophoresis diagram of ITS1 gene amplicons.

M, marker; +, positive control (fungus Ophiocordyceps sinensis); 1, L.LK.4L; 2, L.LK.06; 3, L.LK.07; 4, L.LK.13; 5, L.LK16; 6, L.LK.17; 7, L.LK.18; 8, L.LK.19; 9, L.LK.20; 10, L.LK.21; 11, L.LK.22; −, negative control (water).

Detection of ranavirus

For primer set RanaM68F/ BIVMCP154, only samples L.LK.17 and L.LK.20 showed a weak band at the expected size of ranavirus MCP gene (Fig. 8A). For primer set RanaJP556F/ RanaJP 772R, no expected sized amplicon was found in all samples (Fig. 8B). For primer set OIT1/ OIT2R, only L.LK.13 showed a band at the expected size of ranavirus MCP gene (Fig. 8C). Therefore, no ranaviruses were considered positively. Besides, no viruses were observed under SEM.

Figure 8 Agarose gel (2%) electrophoresis diagram of ranavirus specific MCP gene amplicons.

(A) Amplicons of primer set RanaM68F/ BIVMCP154. The expected size was 230 bp. M, marker; 1, L.LK.16; 2, L.LK.17; 3, L.LK.19; 4, L.LK.20; 5, L.LK21; 6, L.LK.22; 7, L.LK.07; 8, L.LK.13. (B) Amplicons of primer set RanaJP556F/ RanaJP 772R. The expected size was 217 bp. M, marker; 1, L.LK.16; 2, L.LK.17; 3, L.LK.19; 4, L.LK.20; 5, L.LK21; 6, L.LK.22; 7, L.LK.07; 8, L.LK.13; 9, L.LK.18. (C) Amplicons of primer set OIT1/ OIT2R. The expected size was 500 bp. M, marker; 1, L.LK.16; 2, L.LK.17; 3, L.LK.19; 4, L.LK.20; 5, L.LK21; 6, L.LK.22; 7, L.LK.07; 8, L.LK.13.

Confirmation of pathogenicity of A. chelonae

To verify the pathogenicity of A. chelonae, we isolated and purified the bacteria from nodules resected from crocodile lizard skin using Columbia blood agar base plates, and a pure A. chelonae culture (strain LK16-18) was obtained. This bacterium was filamentous under light microscopy and SEM. Colonies of A. chelonae on Columbia blood agar base plates were beta hemolytic, rough, adherent, and white (Fig. 9).

Figure 9 Beta hemolysis of Austwickia chelonae on a Columbia blood agar base plate.

Half a month after hypodermic inoculation with A. chelonae, growing nodules were found on the inoculated sites of the tested Chinese skinks, as expected (Fig. 10). As the nodules growing, the experimental Chinese skinks appeared to eat less or did not eat. In addition, they had difficulty in moving. Moreover, all of the Chinese skinks inoculated with A. chelonae died in 21–44 days. The 16S rDNA sequencing results showed that the bacteria in all nodules resected from dead Chinese skinks were A. chelonae. Furthermore, A. chelonae colonies were re-isolated from the Chinese skink nodules at Columbia blood agar base plates. Filamentous bacteria were observed at H&E staining micrographs of Chinese skink nodules. Necropsy of the dead Chinese skinks showed that small white pellets were found at the surface of visceral organs. The 16S rDNA sequencing results showed that these pellets were bacterial colonies of A. chelonae. These results indicated that A. chelonae had spread into viscera.

Figure 10 Chinese skink hypodermically inoculated with Austwickia chelonae (photo time: 44 days after inoculation).

Arrows note the visible nodules on the left side compared with the right side. Photo credit: Haiying Jiang.

No nodules were found in the skin of experimental Chinese skinks that inoculated with S. enterica LK18-19, Acinetobacter sp. Exi5-53, P. protegens Exi5-13, and PBS. Two Chinese skinks died in 4–5 days after inoculated with P. protegens Exi5-13. The left limbs were red and swollen, but no nodules were observed at the inoculated sites and internal organs of these two dead animals. The other experimental Chinese skinks were still alive at the end of the experiment.

All the above results led to the conclusion that A. chelonae caused the nodules in crocodile lizards reported in this study.

Source of A. chelonae

To trace the source of the pathogen six soil samples and two water samples from the crocodile lizard living environment were sequenced. The results showed that A. chelonae was found in some soil and water samples with very low relative abundances (Fig. 6, Table S1).

Sensitivities of A. chelonae to antibiotics

Austwickia chelonae was tested against twelve antibiotics to find effective drugs to control this disease in crocodile lizards (Table 3). The most sensitive antibiotic was cephalothin, followed by ampicillin and minocycline. Furthermore, A. chelonae was also sensitive to levofloxacin, rifampicin, erythromycin, ciprofloxacin, and piperacillin. However, this bacterium was not sensitive to kanamycin, gentamicin, streptomycin or clarithromycin.

Discussion

Declines in crocodile lizard populations continue despite efforts to alleviate the situation. While conservation efforts are currently focused on habitat protection, disease research is an important issue that needs to be solved for all types of wildlife, especially for endangered animals, such as crocodile lizards, because the disease may significantly increase the risk of extinction for endangered animals (Berger et al., 1998; Daszak, Cunningham & Hyatt, 2000; Hellebuyck et al., 2017; Schumacher, 2006). For example, chytridiomycosis is now recognized as a driver of amphibian population declines (Berger et al., 1998; Fisher, Garner & Walker, 2009). Many nature reserves or zoos treat or breed threatened species with the aim to reintroduce these individuals to their natural habitats. Disease research may improve survival success in these projects, prevent disease vectored into the wild population or other populations following reintroduction, and therefore prevent species extinction.

Both infectious and noninfectious agents can cause dermatosis in lizards. Some primary pathogens that fulfill Koch’s postulates have been described; for example, Chrysosporium anamorph of Nannizziopsis vriesii (CANV) is the etiological agent of “yellow fungus disease” in veiled chameleons (Chamaeleo calyptratus) (Paré et al., 2006), and Devrisea agamarum causes dermatitis in agamid lizards and Lesser Antillean iguana (Iguana delicatissima) with skin lesions (Hellebuyck, Martel & Chiers, 2009; Hellebuyck et al., 2017). Secondary infections with bacteria or dermatomycosis are common in lizards (Hellebuyck et al., 2012; Mader, 2006). Parasites, algae and neoplasms can also affect the skin (Hernandez-Divers & Garner, 2003; Mader, 2006; Van As et al., 2016).

In this study, we report the application of metataxonomics in the pathogenic identification of the cutaneous granuloma disease in crocodile lizards, representing the first report of infectious disease in crocodile lizards. For fungal infection analysis, not all nodules contained fungi, which is inconsistent with Koch’s postulates. The fungus was not observed in the nodules stained with GMS stains. Therefore, it was concluded that cutaneous granuloma in crocodile lizard was not caused by fungi. For bacterial infection analysis, metataxonomics quickly narrowed the candidate pathogens to just five bacteria (A. chelonae, Salmonella sp., Acinetobacter sp., Pseudomonas sp. and Halomonas sp.) through flower plot and related abundance analysis. Combined with the filamentous feature of microorganisms observed under H&E staining, GMS staining and SEM, the pathogen (A. chelonae) was identified rapidly and successfully. Pure A. chelonae isolated from the nodule was similar to the filamentous bacterium observed by SEM and could induce cutaneous nodules in Chinese skinks (Fig. 10). The other three common bacteria, Salmonella sp., Acinetobacter sp., Pseudomonas sp. did not induce nodule in the experimental Chinese skinks. Taken together, these results well revealed that the skin nodules on the crocodile lizards were caused by A. chelonae. As the presence of heart and gastrointestinal nodules in crocodile lizards and the presence of A. chelonae in Chinese skink viscera, the authors speculated that the death of crocodile lizards was a result of A. chelonae spreading into internal organs. The other bacteria may serve as a secondary infection which also contributes to the death of lizards. For example, the predominant bacteria Salmonella, Fusobacterium, Bacteroides, Chryseobacterium and Morganella are pathogens or conditional pathogens identified in other animals. Secondary infected by these pathogenic or conditional pathogenic bacteria may aggravate the disease situation in crocodile lizards. Although Halomonas sp. was common in the nodules, it had low abundance in nodules and was not isolated in this study. Therefore, Halomonas sp. was not inoculated into Chinese skinks.

Austwickia chelonae is a filamentous, Gram-positive Actinobacterium, which was named Dermatophilus chelonae in the original reference and recently reclassified as a new genus within the Dermatophilaceae family (Hamada et al., 2010). The disease resulting from A. chelonae infection is called dermatophilosis. Dermatophilosis has been reported in a number of vertebrates, including a variety of mammals (Aubin et al., 2016; Caron et al., 2018; Gebreyohannes, 2013; Lunn et al., 2016; Nemeth et al., 2014), birds (Scaglione et al., 2016; Shearnbochsler et al., 2018), reptiles (Hellebuyck et al., 2012; Tamukai et al., 2016; Wellehan et al., 2004) and humans (Amor et al., 2011; Aubin et al., 2016; Burd et al., 2007). This disease is an important zoonotic skin disease in domestic animals leads to significant economic losses (Ndhlovu & Masika, 2016; Shaibu et al., 2010). Symptoms of dermatophilosis in reptiles include surface crusts, necrotic cellular debris, inflammatory cells, nodular hyperkeratosis, necrosis of the epidermis, and caseous subcutaneous abscessation (Bemis, Patton & Ramsay, 1999; Mader, 2006). The mainly causative agent of dermatophilosis is D. congolensis. Dermatophilosis resulted from A. chelonae infection was initially reported in a nose scab on a snapping turtle in Australia (Masters et al., 1995). Austwickia chelonae infection was also detected in king cobra (Ophiophagus hannah) (Wellehan et al., 2004). In lizards, an outbreak of A. chelonae co-infected with ranavirus infection was previously reported in inland bearded dragons (Pogona vitticeps) in Japan (Tamukai et al., 2016). In this study, ranavirus was not detected, but every A. chelonae infected sample was found to be co-infected with Salmonella sp., Acinetobacter sp., Pseudomonas sp. and Halomonas sp. In addition, the experimental infection also occurred in Chinese skinks (Fig. 10). Besides reptiles, A. chelonae was also detected in free-living hooded crows (Corvus corone cornix), a bird species that had proliferative and crusted foot lesions (Scaglione et al., 2016). Moreover, A. chelonae also produced dermatophilosis lesions onto sheep, rabbits and guinea pigs after inoculation (Masters et al., 1995).

For the treatment modalities of A. chelonae infection, surgical resection of the lesions as early as possible and supplementation with antibiotics were recommended. According to the previous study, A. chelonae was susceptible to penicillin, tetracycline, chloramphenicol and sulfafurazol, and resistant to polymyxin, streptomycin and neomycin (Masters et al., 1995). This study revealed that A. chelonae was also susceptible to cephalothin, minocycline, levofloxacin, rifampicin, erythromycin, ciprofloxacin and piperacillin while resistant to kanamycin, gentamicin and clarithromycin. Compared with A. chelonae and D. congolensis, most information about antibiotics susceptibility are same. However, D. congolensis was resistant to levofloxacin and susceptible to gentamicin (Amor et al., 2011).

For the pathogenic source, A. chelonae was found in some water and soil samples collected on site at the same time, but not detected in gut microbiomes of crocodile lizards (Jiang et al., 2017). Therefore, A. chelonae may come from soil or water in the living environment. In that case, environmental disinfection would be an effective preventive method for this disease. This disease has seldom happened nowadays after regularly disinfected the ecological simulation pools using KMnO4. However, further investigation of the exact source of A. chelonae is needed, such as the bacteria from food, normal skin microbiota and bacterial flora of the local soil and water. For the mode of A. chelonae transmission to crocodile lizards, it is hypothesized that this infection might be caused initially by local trauma, followed by the invasion of A. chelonae and other secondary infections. Because crocodile lizards bite each other when they are competing for foods, territory and mates, they are prone to suffering trauma. The most common factors contributing to the pathogenesis of dermatophilosis are skin trauma, prolonged wetting, high humidity, high temperature and concurrent diseases (Gebreyohannes, 2013). In addition, pathogen transmission is also affected by environmental stress (e.g., relative overcrowding, habitat migration), climatic conditions, seasonal changes and diets (e.g., food availability and diversity) (Ryser-Degiorgis, 2013).

Conclusion

This study is the first to describe, identify and isolate Austwickia chelonae as the primary pathogen underlying cutaneous granulomas in crocodile lizards and Koch’s postulates were fulfilled using Chinese skinks. Secondly, our study highlights the role of potential co-infections of A. chelonae with other bacteria, such as Salmonella, Fusobacterium, Bacteroides, Chryseobacterium and Morganella, in crocodile lizard dermatophilosis. Thirdly, A. chelonae was proved to be sensitive to cephalothin, minocycline, ampicillin, levofloxacin, rifampicin, erythromycin, ciprofloxacin and piperacillin but resistant to kanamycin, gentamicin, streptomycin, and clarithromycin. It is noteworthy that our research indicated that the application of metataxonomics was effective in the identification and diagnosis of pathogens in wildlife animals. Metataxonomics can reduce the turn-around time and provide accurate identification than the conventional culture method. This is a boon for wildlife, whose diseases and pathogens are poorly understood compared to domestic animals.

Supplemental Information

Table S1 Relative abundance of each OTU

Click here for additional data file.

Figure S1 Numbers of effective sequences and OTUs in each sample

Total tags (red, indicates effective tags) were sequences without low-quality sequences and chimeras and were used for annotation and other analyses. Taxon tags (blue) represent the sequences that could be clustered into OTUs and annotated. Unclassified tags (green) refer to tags without annotated information. Unique tags (orange) refer to tags that occurred only once and could not be clustered into the number of OTU tags.

Click here for additional data file.

Figure S2 Rarefaction curves of each sample

Click here for additional data file.

Figure S3 Phylogenetic relationship of top 100 genera of bacteria

The branches were colored by phyla. The bars out of the branches represent relative abundances and colored by samples.

Click here for additional data file.

The authors thank Prof. Xudong Cao and Dr. Hafiz Ishfaq Ahmad for their critical reading, advices and language modification. Our thanks also go to Dr. Lv-Ping Zhang for his help for SEM microscopy.

Additional Information and Declarations

Competing Interests

Author Contributions

Animal Ethics

Data Availability

The authors declare there are no competing interests.

Haiying Jiang conceived and designed the experiments, performed the experiments, analyzed the data, prepared figures and/or tables, authored or reviewed drafts of the paper, approved the final draft.

Xiujuan Zhang performed the experiments, analyzed the data, contributed reagents/materials/analysis tools, approved the final draft.

Linmiao Li performed the experiments, contributed reagents/materials/analysis tools, approved the final draft.

Jinge Ma, Huiming Li and Zhengjun Wu contributed reagents/materials/analysis tools, approved the final draft.

Nan He and Haiyang Liu contributed reagents/materials/analysis tools, prepared figures and/or tables, approved the final draft.

Richou Han authored or reviewed drafts of the paper, approved the final draft.

Jinping Chen conceived and designed the experiments, authored or reviewed drafts of the paper, approved the final draft.

The following information was supplied relating to ethical approvals (i.e., approving body and any reference numbers):

All experimental animal procedures in this study were approved by the Committee on the Ethics of Animal Experiments of the Guangdong Institute of Applied Biological Resources (GIABR180928 and GIABR201027) and followed basic principles.

The following information was supplied regarding data availability:

All raw sequences obtained from high-throughput sequencing are available in the NCBI Sequence Read Archive (SRA) under the accession number SRP152217. The complete genome sequence of A. chelonae LK16-18 is available in the NCBI GenBank and published (Jiang et al., 2018). The 16S rRNA gene sequences of A. chelonae LK16-18, S. enterica LK18-19, Acinetobacter sp. Exi5-53 and P. protegens Exi5-13 are available in the NCBI NR database under the accession numbers MK110377, MK235186, MK235211 and MK235212.

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
