# Peer review of "Identification of Austwickia chelonae as cause of cutaneous granuloma in endangered crocodile lizards using metataxonomics"

_PeerJ, doi:10.7717/peerj.6574_

## Round 0.1 · original submission · Major Revisions

Please take special care about the referees' comments regarding the sampling, experimental design and description of methods; other remarkable points to consider are about the putative presence of other pathogens that could have contributed for the death of the affected animals.

·

Basic reporting

The manuscript #31617 "Rapid identification of cutaneous granuloma pathogen that causes many deaths in crocodile lizards using metataxonomic next-generation sequencing" describes identification of Austwickia chelonae as pathogen for cutaneous granulomas in crocodile lizards. Experimentally infection was performed in Chinese skinks for confirmation of pathogenicity. The manuscript is well referenced and important for reptile medicine as well as conservation issues at least in crocodile lizards. Abstract and introduction is well written and later one provide a good background for the presented content. Structure of the manuscript is conforming with the journal standard, but needs improvement by reorder of the material and methods part as well as using of subheadings in the result part. Raw data are shared mostly, but Sequence of strain LK16-18 have to be stored in the NCBI database and Acc. Nr. have to be given.

Major concerns:
Material and Methods:
Line 102-105: As the lesions and animals, which suffered on this lesions, comes first before diagnostic methods, it should appear at first. Therefore, a subheading would be useful, like: Animals and sampling.
Please give coordinates of the Guangdong Luokeng Shinisaurus crocodilurus National Nature Reserve.
Please give a comment if the lizards were free living ones or not. If they were free living ones, give the capture method. Where do you samples the animals? I assume you sampled them in-house, as you stored the obtained samples immediately in liquid nitrogen.
Give age of the lizards, or at least if all were adult ones.
Please stated clearly under this subheading how many animals were sampled. In total 11 granulomas were sampled but it is not clear, how many animals were sampled, even not in table 1. Reference to Fig. 1 is not helpful, as eight locations of the granulomas are shown. Please add here also, if you have stored samples obtained from one animal different, as you mentioned later bacteriological cultivation, histopathology and scanning electron microscopy.
Line 140-146: This part needs more explanations and should be the last paragraph inside material and methods part.
How many skinks were infected, age of the animals, were they tested before infection? How long was the experimental period? What happens after the experimental part at the end with the skinks? How do you re-isolate the pathogen? Did you perform histopathology to obtain pathomophological correlate? Please insert this paragraph after Bacterial isolation and cultivation.


Results:
Subheading are needed for structure.
Line 191-192: This sentence is a method and should be added and explained more in the material and method part.
Line 194-196 have to be placed and discussed in the discussion part.
Line 201-204: This sentence has to be placed in the method part under subheading “artificial infection”

Figures:
Fig. 1: Delete or change number B, C, D and E, as animals were held without gloves on this photographs. As the zoonotic risk is discussed and at time point of sampling the outcome is unclear, it is not adequate to handle reptiles with skin lesions without gloves – biosecurity! Delete the term “symptoms and”. Please give explanation in the caption for labelling with A, B, C….
Fig. 2: Change or delete this micorphotograph. Use a lesser magnification, so that the whole lesion is shown. I assume a fibrinous granuloma, but it is not shown on the used micriphotograph. Grocott-Gomori's methenamine silver stain of the lesions would be useful to give an adequate histopathological correlate.

Tables
Table 2:
Definition of sensitivity grade is unclear. Please provide measurable data for replication of your results.

References:
References in the text have to be ordered chronologically. DOI numbers have to be included in the reference list.

Experimental design

Content of the manuscript is a veterinary one about a disease and finding of the etiological pathogen, which seems not to be covered by the Aims and Scope of the Journal.
Cause of cutaneous lesions in crocodile lizards has not described before, which has been identified as a knowledge gap. The diagnostic work up is interesting, advanced and an ethical statement is given, but the method part need much more attention for replication as well as for approving the robustness of given data.

Major concerns:
Material and Methods:
Line 147-150: What did you mean with “round papers containing antibiotics”? If you used standardized disk diffusion test, please state so and give the content and provider for the disks correctly. How many isolates were tested? Give the definition you used for sensitivity.

Validity of the findings

Major concerns:
Results:
Line 191-196: As you have not performed histopathology including PAS-reaction or Grocott-Gomori's methenamine silver stain in all 11 nodules to prove your molecular biological results, you can´t give such conclusion.
Line 199: Sequence of strain LK16-18 have to be stored in the NCBI database and Acc. Nr. have to be given here.

Discussion:
Line 246-253: Most of the given references in mamals and humans are for D. congolensis, therefore it is not useful mix both pathogens here. Please indicate clearly, in which species (excl. reptiles) infection with A. chelonae has described so far. To my best knowledge A. chelonae has only be isolated from reptiles so far, so that a zoonotic risk is unlikely according to the scientific knowledge.
Line 285-286: Co-infection with ranavirus seems to be very interesting, beside skin traumata, for formal pathogenesis of the disease. Why didn’t you examine your lizards for ranavirus to exclude this etiology? Please discuss this issue here.

Additional comments

Minor comments:
Title:
Change to: Identification of Austwickia chelonae as cause of cutaneous granulomas in crocodile lizards (Shinisaurus crocodilurus)
Material and Methods:
Line 104 and 106: delete the term “epidemic material”, as you sampled skin lesions or skin nodules
Line 106-108: Could you describe the composition of the soil sample and the origin of the water sample.
Line 130-132: add cultivation time
Line 142: change to Plestiodon chinensis
Line 152: delete “epidemic materials”, see above
Line 153-156: Grocott-Gomori's methenamine silver stain of the lesions would be useful to give a histopathological correlate.

Results:
Line 162-163: Delete this sentence, as already mentioned in the material and method part.
Line 169-170: Describe or name the cells you have seen.
Line 184-185: Delete sentence, as Koch´s postulates are well known to the readers.

Discussion:
Line 223: Please give a reference for the sentence: “Dermatosis…in lizards”, or delete this sentence.
Line 289-293: see above, include this part into the discussion above

Reviewer 2 ·

Basic reporting

Spelling and grammar as well as structure should largely improve. References are missing in certain sections.

Experimental design

Especially the protocol for antimicrobial susceptibility testing is missing as well as the experimental inoculation protocol. Many other aspects are missing in the M&M and Results section as indicated below.

Validity of the findings

Certain inappropriate or invalid conclusions are made or should be nuanced 'see further remarks).

Additional comments

Few reports document the occurrence of a bacterial disease posing a conservation treat to a wild living, endangered Squamate species. Although this article provides valuable information this article should be thoroughly reworked.
Some general and more specific remarks are made below.

Title
The formulation of the title is awkward. Please use scientific terminology. What was the main goal of the study; revealing the causative agent of the observed disease or developing a rapid identification strategy? Why does it seem as rapid identification is the main emphasis, while the unravelling the role of the bacterium as a primary or facultative pathogen causing high morbidity in a critically endangered species seems to be more important?

Abstract
The abstract should be rewritten. It is unclear if the authors are talking about wild, rescued lizards or lizards that are part of a breeding/conservation program. What part of the wild population is affected (quantitatively, geographically,…)? The authors claim that serious diseases affect the population but eventually focus on a single clinical entity caused associated with a single agent.
Conclusions should be formulated appropriately: did the authors try to fulfill Koch’s Postulates? Is A. chelonae considered as a primary pathogen or a facultative pathogen? The use of antibiotics would only be a part of a control strategy. Is this really considered as a cure, a solution for this problem? Having performed antimicrobial susceptibility testing does not necessarily mean that the use of antimicrobials will be achievable or efficient in the field… What about the origin of the disease and the factors promoting it?

Introduction
L60: this sentence seems to be redundant
L61-64: reformulate. Clearly state the impact, occurrence of the observed disease. What are the other skin diseases besides the granulomas?
Did the lizards die because of the skin disease and why (also see further)? Did the disease appear in captive bred animals, wild caught animals? Was this also seen in wild animals?
L66-67: this seems to be a different clinical presentation. Did the authors sample these lesions?? Was A. chelonae also isolated and if yes, was it related to the other isolates obtained in the other site? If not, this information should be excluded or only mentioned in the discussion. In other words, did the authors investigate these animals?
The authors should explain what was found during necropsy of affected animals and how a dermal disease led to the death of affected lizards. What was the pathogenesis/pahtology? Please provide results of cultures (including mycology) performed on internal organs as well as histology. Was acid fast staining performed?
L71-72: Again, please check the entire text for grammar and spelling. Does this conclusion belong in the introduction? This needs to be discussed in the discussion section. A control strategy taking into account all factors that promote disease and allow the bacterium to have such an impact should be proposed.
L73: this should not necessarily be a general statement. Not all infectious agents can be cultured. The authors already assume that the observed skin disease has a bacterial origin in the introduction section while e.g. viral, mycotic parasitic and even non-infectious causes, environmental should be considered. For this reason the next paragraph should not(necessarily) be part of the Introduction.
L78: fertile? L84: noisy?
L90-95: please formulate the primary objectives of the study

M&M
This section should be ordered more logically. All materials and methods are focused on A. chelonae isolation and identification but at this stage all possible causes should have been considered and all methods used to screen or exclude these should be described here and in the Results section. Mycology, virology?
L105: 96 lizards showed the disease but only 11 were sampled. See previous remark: only lizards from one site were sampled? Why? Please describe full necropsy results. Other testing should have been performed as mentioned above. Please provide the results.
L140: inoculation? Why were Chinese skinks used? Why was this inoculation dose used? Why was this inoculation route used? The entire follow-up period should be described: what signs were looked for, how many times a day, what were the housing conditions? Were animals/lesions sampled post inoculation? Etc.
L141: please clarify candidate pathogen
L151: Histology and SEM? L154: is this the most appropriate stain for A. chelonae?
What protocol was used for susceptibility testing and why? What are the limitations in comparison to another methods? Can the others draw conclusions based on a limited number of isolateS?

Results
L162: please provide numbers throughout time
L169: was acid fast stain, Gram staining performed?
The authors should discuss the importance of the cultured bacteria in the discussion in the light of known pathogenicity in reptiles/lizards based on the available literature: what agents were considered as contaminants vs pathogens.
It is quite unusual to have such elaborate mixed infection in a single skin lesion. The owners really need to explain this. Why was A. chelonae eventually selected? If they would have inoculated most of the other bacteria at the same dose subcutaneously diseases would also have developed I guess…? Please clarify.
Did the isolates fully correspond to A. chelonae based on sequencing. Were variations observed?
L175-176: What are the authors trying to state?
L186: the context seems to be missing? The criteria towards fulfilling Koch’s postulates should be explained clearer, how these were met and especially the reason why the authors chose A. chelonae as the main causative agent of the observed nodules. What is the role of the other bacteria that occurred in all lesions? Many of this will also have caused granulomas/abscesses if inoculated at the same dose? Why granulomas and not abscesses?
The main argument seems to be the SEM results. The histological and SEM findings should be described better. How were these samples processed to assess the importance of the other 4 bacteria that were found in all nodules?
L195-196: results of histology (was PAS stain performed) and culture should be included, otherwise this is too speculative.
L199-201: context?
L203: another seems to contradict the previous sentence. Why exactly Chinese skink? Again, the experimental inoculation protocol seems to be missing in the M&M and Results section: how many animals, how was follow-up performed, how many animal (!), general disease signs developed etc. How was the inoculum prepared? What isolate was used (type strain submitted)? How was the used strain related to the other isolates from the crocodile lizards and to other type strains of closely related Actinobacteria?
L207-208: please rephrase
L209: is the soil considered as the main source? What about other substrates? Can animals be asympotmatic carriers?

Discussion
This section should be rewritten and reordered. Check for correct referencing.
L219-222: this paragraph is poorly written. Please rewrite and certainly provide references (what literature data are available of outbreaks of infectious diseases with an impact (endangered ) in free ranging lizard populations or reptiles in general)?
L223: what does this sentence mean? References? L227: only agamids? Recent outbreak in wild iguanas has been published?
L233: see previous remarks? Do the authors consider this as a sound scientific approach?
L241-242/244-245: can this really be put forward as a conclusion? Specific for wildlife rather than e.g. sample type?
L246: Why did the crocodile lizards succumb to localized granulomas taking into account the literature described here? L260: supplementation What about the zoonotic potential
L268: Formulating an opinion about the route of infection and the reason why the lizards are susceptible to developing A. chelonae associated disease is highly important. Is the bacterium new in the lizards’ environment? Why have they become susceptible to wound infection? What research is needed to unravel this?
L282: What are the authors tring to say? Is A. chelonae a primary pathogen? What is the true importance?

Conclusion: please reformulate based on the previous remarks

Figure captions: the information provided should be provided more elaborately
Figure 1
Why do the lizards succumb to these relatively small skin lesions? Any proof for systemic infection (see previous remarks)?
Figure 6 and 9: relevance?

---

## Round 0.2 · Minor Revisions

The corrections and changes pointed by the referees were made accordingly. Only minor corrections in English and grammar are needed.

·

Basic reporting

All points raised in the initial review has answered and corrected sufficiently. English is clear, references are sufficient, structure of the article incl. figs and tables presented understandable to follow the results and the hypothesis.

Experimental design

All points raised in the initial review has answered and corrected sufficiently. Identification of Austwickia chelonae as pathogen for cutaneous granulomas in crocodile lizards has been described. Experimentally infection was performed in Chinese skinks for confirmation of pathogenicity. The manuscript is well referenced and important for reptile medicine as well as conservation issues at least in crocodile lizards. Cause of cutaneous lesions in crocodile lizards has not described before, which has been identified as a knowledge gap. The diagnostic work up is interesting, advanced and an ethical statement is given. Method part has been improved, so that replication seems possible and underlines the robustness of given data.

Validity of the findings

Method part has been improved, so that replication seems possible and underlines the robustness of given data. Conclusion are well stated and linked to original research question.

Additional comments

Well done, thank you for corrections and answering all raised questions by the reviewers.

Reviewer 2 ·

Basic reporting

The structure of the article has improved by addressing the reviewers' comments. Spelling and grammar as well as structure should still improve. Several parts of the text can be formulated in a more scientific way.

Experimental design

The authors have done a good job by providing lacking information in the M&M and results section.

Validity of the findings

Certain statements and conclusions could be written more appropriately or can be nuanced more. See suggestions below.

Additional comments

Although the authors have made great effort to address the comments of the reviewers and the manuscript has improved some concerns remain:
- I think that the formulation/grammar of the title can still be improved.
- In my modest opinion, grammar and spelling should still largely improve throughout the manuscript, especially in the abstract, introduction and discussion section. The authors do not always explain why they choose not to rephrase parts of the texts.
Several parts of the text can be written more concisely and clearer. E.g. in the abstract confusing information is provided (‘…have been increasingly infected by serious diseases, which cause many deaths in each year, especially the rescued and the breeding individuals…. However, pathogens underlying these diseases are unclear.’ contradicts the introduction ‘…, infectious pathogens in crocodile lizards have not been previously reported (Brady et al. 2016). ‘. The reader may assume that many (infectious) diseases are known in free ranging crocodile lizards, while they are mainly threatened by human influences. This manuscript details the occurrence of granulomatous skin disease in rescued/breeding animals by A. chelonae in ecological simulation pools. Could the disease be considered as a management related facultative/ opportunistic disease…? To give just one example: the authors correctly state that wound infection due to e.g. biting lesions may be an important facilitating factor but I assume that this facilitating factor is mainly important in the ecological simulation pools. Biting behavior is probably also common in wild individuals but relative overcrowding, high infection pressure, concurrent disease, immunosuppression, semi-natural conditions etc. is what might really enable the bacterium to cause largescale and persistent disease (after all the natural biting behavior in this species is there longer. Although the authors attempt to clarify this to a certain degree in the discussion, clarification of the occurrence of the disease and explaining why the bacterium causes the observed problem/disease could still be improved throughout the manuscript.
- Is surgical excision and antimicrobial treatment considered as a realistic option? How will elimination of (subclinical) infection be confirmed? Following the identification of this pathogen as a cause of granulomatous dermatitis it may be considered that the focus should be on unravelling the facilitating factors that allow the bacterium to cause clinical infection, studying the possible presence of asymptomatic carriers and environmental persistency to be able to control this problem in the reserves. Should the authors focus more on the fact that this problem would not necessarily occur in a healthy, free-ranging population and that the disease may be vectored into the wild/other populations following (re)introduction from the affected population(s)?
- The authors could try to reformulate several parts of the text. A few examples:
Introduction: ‘However, during the work of rescue and breeding, the crocodile lizards are prone to serious diseases that cause many deaths each year (Jiang et al. 2017). The most frequently disease is skin diseases.’ Please correct for grammar. Why would the lizards be prone to the work of rescue to serious diseases? What (skin) diseases besides the one that is described in this paper?
‘However, disease diagnosis of wildlife animals can be problematic due to lack of information regarding pathogens that cause these diseases.’ Why would identifying the cause of an enigmatic skin disease in wild animals be more challenging then in animals in a breeding program or even in captive animals?
Discussion: ‘While conservation efforts are currently focused on habitat protection, disease research is a problem that needs to be solved for all types of wildlife, especially for endangered animals, such as crocodile lizards, because the disease may significantly increase the risk of extinction for endangered animals’. Is disease research a problem? The authors may consider to write statements like this in a more concise and clear way.
- The Conclusion section still does not seem to contain the major, most relevant conclusions from this study

---

## Round 0.3 · accepted · Accept

The corrections were made accordingly.

#